# TFF3 interacts with LINGO2 to regulate EGFR activation for protection against colitis and gastrointestinal helminths

Nicole Maloney Belle[1], Yingbiao Ji[1,5], Karl Herbine[1,5], Yun Wei[2,3,5], JoonHyung Park[1,5], Kelly Zullo[1], Li-Yin Hung[1,2], Sriram Srivatsa[1], Tanner Young[1], Taylor Oniskey[2], Christopher Pastore[1], Wildaliz Nieves[4], Ma Somsouk[4] & De'Broski R. Herbert[1,2]*

Intestinal epithelial cells (IEC) have important functions in nutrient absorption, barrier integrity, regeneration, pathogen-sensing, and mucus secretion. Goblet cells are a specialized cell type of IEC that secrete Trefoil factor 3 (TFF3) to regulate mucus viscosity and wound healing, but whether TFF3-responsiveness requires a receptor is unclear. Here, we show that leucine rich repeat receptor and nogo-interacting protein 2 (LINGO2) is essential for TFF3-mediated functions. LINGO2 immunoprecipitates with TFF3, co-localizes with TFF3 on the cell membrane of IEC, and allows TFF3 to block apoptosis. We further show that TFF3-LINGO2 interactions disrupt EGFR-LINGO2 complexes resulting in enhanced EGFR signaling. Excessive basal EGFR activation in Lingo2 deficient mice increases disease severity during colitis and augments immunity against helminth infection. Conversely, TFF3 deficiency reduces helminth immunity. Thus, TFF3-LINGO2 interactions de-repress inhibitory LINGO2-EGFR complexes, allowing TFF3 to drive wound healing and immunity.

[1] Department of Pathobiology, University of Pennsylvania School of Veterinary Medicine, Philadelphia, PA 19140, USA. [2] Division of Experimental Medicine, University of California, San Francisco, San Francisco, CA 94110, USA. [3] Department of Inflammation and Oncology, Amgen Inc., 1120 Veterans Boulevard, South San Francisco, CA 94080, USA. [4] Division of Gastroenterology at ZSFG, University of California, San Francisco, San Francisco, CA 94110, USA. [5] These authors contributed equally: Yingbiao Ji, Karl Herbine, Yun Wei, JoonHyung Park. *email: debroski@vet.upenn.edu

Gastrointestinal (GI) tissue undergoes regenerative and injury-induced repair through diverse pathways that span arachidonic acid metabolites (e.g., prostaglandins, leukotrienes), growth factors (e.g., KGF, EGF), regulators of development (e.g., Notch, Wnts, Hippo), and reparative cytokines (e.g., Trefoil factor family proteins (TFF1-3))[1–8]. TFF3 is a small (6 kDa) secreted glycoprotein produced by goblet cells that drives wound healing throughout the respiratory, ocular, biliary, genitourinary and GI mucosa[9–11]. Despite identification over 30 years ago, and numerous studies implicating TFF3 in human diseases, the existence and/or identity of a bona-fide receptor responsible for TFF3-mediated cellular migration, anti-apoptotic activity, cell adhesion control, and/or immunoregulation remains unknown[12–15]. Autoradiography and chemical cross linking-based studies have identified TFF binding sites along the GI tract of rodents and most data suggests that TFFs can induce MAPK, beta catenin, and/or EGFR activation[16–19]. Still, whether a TFF3 receptor exists has been controversial because of evidence that TFF3 (along with TFF2) increase the rheological properties of mucus[20,21]. Given the implication for TFF3 (and TFF2) as a regulator of GI diseases such as Ulcerative colitis and Crohn's disease, there is a clear need for a better understanding of TFF biology.

Evidence shows that pathways of tissue repair/regeneration and immunity functionally intersect[22]. Although TFF3 was previously considered solely a regulator of re-epithelialization, TFF3 has been reported to exert both pro and anti-inflammatory activities[23–27]. TFF3 undergoes auto-induction at its promoter, which is dependent upon STAT3 activity[28] and TFF3 treatment can induce both pSTAT3 and pEGFR activation[14,29], but there are no data that show either gp130 or EGFR directly bind to TFF3[30]. EGFR activation serves an important role in host immunity, particularly against GI helminths such as the rodent hookworms *Heligosomoides polygyrus bakeri* and *Nippostrongylus brasiliensis* (*N.b.*). Amphiregulin, an EGFR ligand, promotes Type 2 inflammation and worm expulsion through EGFR dependent signals[31]. Type 2 immune responses can be synergistic with tissue repair, therefore we reasoned that the reparative and immunoregulatory effects of TFF3 would be evident during helminth infection or colitic inflammation.

This work ventures into the major gap of Trefoil factor family biology by searching for a TFF3 receptor. An approach was used called TRICEPS that employs ligand-guided covalent binding and capture of glycoproteins on the cell surface to identify low-affinity ligand-receptor interactions[32]. This approach led us to discover Leucine-rich repeat and immunoglobulin-like domain-containing nogo receptor-interacting protein 2 (LINGO2), as a critical component of TFF3 cellular responsiveness. LINGO2 is required for TFF3-mediated EGFR and STAT3 activation in intestinal epithelial cells (IEC). Paradoxically, LINGO2 deficiency in mice results in a phenotype opposite of TFF3 deficiency due to repressive EGFR-LINGO2 interactions. In support of this hypothesis, the increased disease severity of DSS colitis in LINGO2 deficient mice and their resistance to *N.b.* infection is both reversed following treatment with the EGFR antagonist Gefitinib. Taken together, these data suggest a model wherein TFF3 binding to LINGO2 reverses tonic inhibition of EGFR by LINGO2, resulting in heightened EGFR activation.

## Results

### LINGO2 expression facilitates responsiveness to TFF3. Anti-inflammatory activities of TFF3 suggested that leukocytes may directly respond to TFF3[25,27]. Given the cell number requirement for the TRICEPS screen ($10^9$/sample) and the apparent lack of success with epithelial cell lines, we used a human macrophage/monocyte cell line (U937) to search for a potential TFF3 receptor.

TFF3-induced interleukin 10 (IL-10) production occurred over several orders of magnitude in these cells, but was lost at higher doses of recombinant human TFF3 (rhTFF3) (Fig. 1a). This bimodal response seemed similar to the activity range for cytokine and growth factor receptors[33], implying that TFF3 was acting through a conventional ligand-receptor interaction.

We postulated that TFF3 interacted with its receptor through low-affinity interactions reliant upon carbohydrates because TFF3 glycosylation has been shown critical for biological activity[34,35], therefore U937 cells were subjected to the TRICEPS™ protocol as a biochemical screening strategy to identify glycosylated transmembrane protein(s) on the cell surface[32,36]. As bait, rhTFF3 was covalently linked to the TRICEPS probe and incubated with PMA-treated U937 that had been treated with sodium periodate. Cell pellets were subjected to glycosidase digestion and peptides generated using mass spectrometry. As a positive control to account for enrichment efficiency, recombinant human insulin was used as bait for the insulin receptor. Whereas the TFF3-probe induced an 8-fold enrichment of LINGO2 (LIGO2) peptide, the insulin-probe induced a 7.76-fold-enrichment in INSR peptide (Fig. 1b). No other enriched peptides in this screen met this level of enrichment or were derived from proteins that satisfied our selection criterion (e.g., extracellular domain, transmembrane region, cytoplasmic tail).

To further explore functional interactions between TFF3 and LINGO2, experiments were performed that employed anti-human LINGO2 Mab treatment of U937. At high doses, anti-LINGO2mAb blocked TFF3-induced IL-10 production (Fig. 1c) and resulted in a dose-dependent neutralization of TFF3-mediated TNF-alpha suppression (Supplementary Fig. 1A). Next, to ask whether LINGO2 and TFF3 were physical binding partners, we performed co-immunoprecipitation experiments utilizing murine TFF3-Fc vs. empty vector Fc incubated with Flag-tagged LINGO2, which revealed that TFF3 could pull down LINGO2 protein (Fig. 1d), indicating that TFF3 could bind LINGO2.

### LINGO2 and TFF3 interact on the epithelial cell surface. Epithelial cell lineages are the major source and target of TFF3[37–39], therefore we focused on whether LINGO2 could capture TFF3 at the epithelial cell surface. HEK293 cells were double transiently transfected with constructs encoding LINGO2-GFP and TFF3-RFP, fixed, permeabilized, and imaged using confocal microscopy. Whereas TFF3-RFP single transfectants had multiple fluorescent puncta throughout the cytoplasm of the positive cells (Fig. 1e), TFF3-RFP transfectants that co-expressed LINGO2-GFP had clear yellow signal at the cell membrane, suggesting co-localization (Fig. 1f and Supplementary Fig. 1B). To confirm the LINGO2 extracellular domain was necessary for the TFF3 interaction, we co-transfected TFF3-RFP with a NH2-terminal LINGO2-Flag truncation mutant (Δ350 AA), which abrogated co-localization and resulted in TFF3 red fluorescent puncta, similar to TFF3-RFP single transfectants (Fig. 1g). These data suggested that the leucine-rich-repeat (LRR) ectodomain region was responsible for ligand binding. As a control and to address reports suggesting that TFF3 may also signal through the SDF-1 receptor CXCR4[40], CXCR4-GFP was co-transfected with TFF3-RFP or SDF1-RFP. Data show TFF3-RFP/CXCR4-GFP transfectants had predominantly red puncta away from the cell membrane suggesting no co-localization, whereas SDF1-RFP transfectants that co-expressed CXCR4-GFP demonstrated yellow signal providing evidence of membrane co-localization (Fig. 1h, i and Supplementary Fig. 1B).

To ensure that co-localization in our double transfection over-expression HEK system was not due to intracellular interactions,

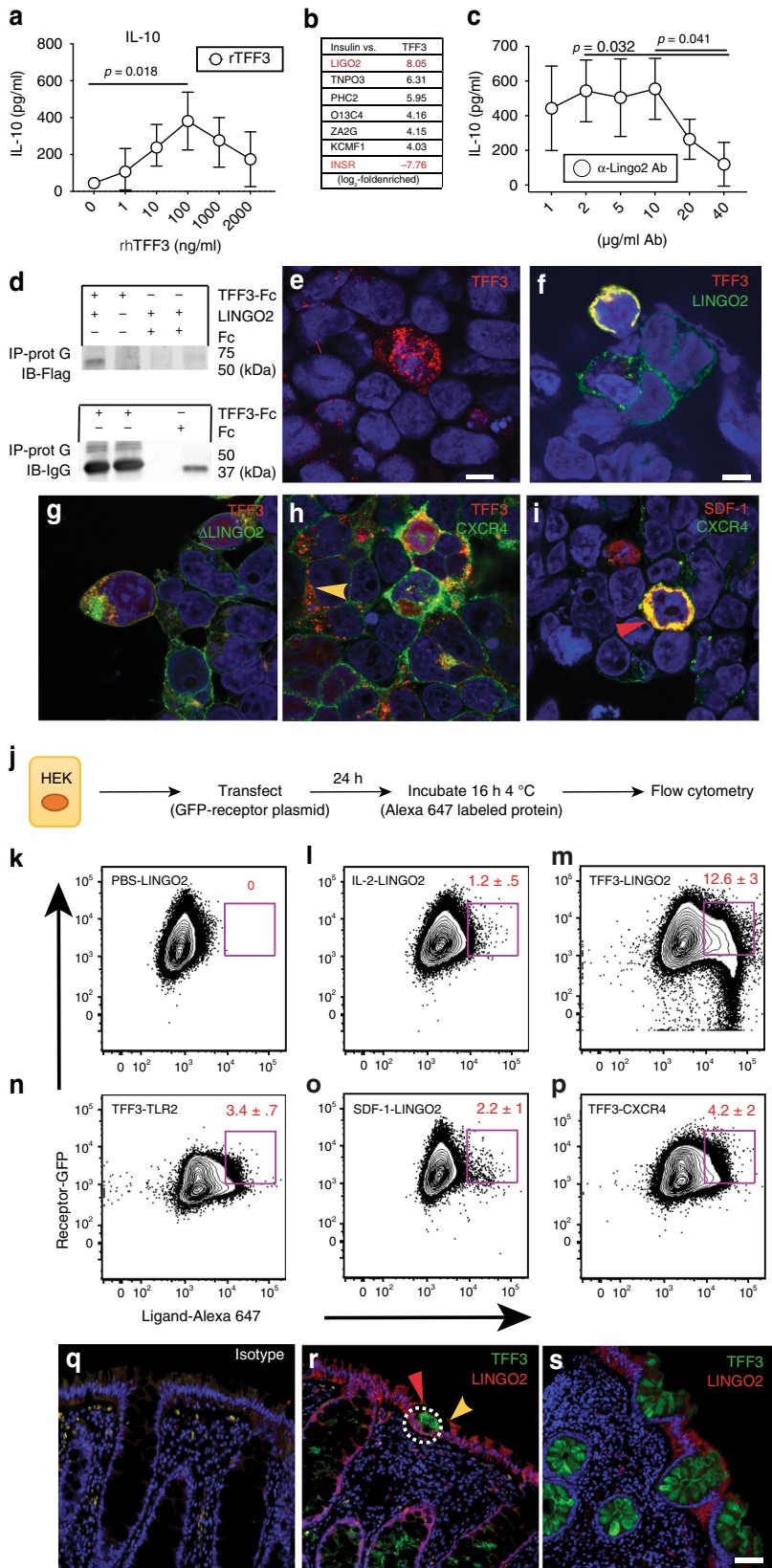

we changed the approach to a flow cytometry strategy involving exposure of LINGO2-GFP transfectants to soluble rTFF3 labeled with Alexa-Fluor 647 (TFF3-647). A flow cytometry gating strategy was devised (Supplementary Fig. 1C) to determine the portion of cells that were doubly fluorescent for GFP and AF-647 (Fig. 1j). Compared to incubation with PBS, or IL-2-647, (Fig. 1k, l) LINGO2-GFP transfectants had a 12-fold increase in double-positive cells following incubation with TFF3-647 (Fig. 1m).

**Fig. 1** Identification of LINGO2 as a putative receptor for TFF3. **a** IL-10 secretion from the human macrophage cell line U937 in response to different rhTFF3 concentrations. Representative of two independent experiments. Means ± SE of quadruplicate wells are shown. **b** Mass spectrometry fold-enrichment peptide score following TRICEPS screen of U937 cells exposed to rhTFF3. **c** IL-10 produced by U937 following overnight treatment with rTFF3 in the presence of anti-human LINGO2 mAb. Means ± SE of quadruplicate wells with results of paired two-tailed *t* test are shown. Representative of two independent experiments. **d** Immunoprecipitation of LINGO2-Flag with either affinity purified TFF3-Fc or Fc only using protein A followed by immunoblotting with anti-Flag Ab (upper blot) or anti-IgG (lower blot). **e** Representative photomicrographs of HEK-293 cells single transfected with TFF3-RFP or **f** co-transfected with TFF3-RFP and LINGO2-GFP vectors, **g** co-transfected with TFF3-RFP and NH$_2$-terminal Flag-LINGO2 truncation mutant (Δ350 AA), **h** co-transfected with TFF3-RFP and CXCR4-GFP or (**i**) CXCR4-GFP and SDF-1-RFP. **j** Schematic of flow cytometry-based strategy to detect doubly fluorescent HEK cells following transfection with GFP labeled receptors and incubation with Alexa 647-labeled soluble protein ligands. **k** Representative flow cytometry dot plots showing the percentage of double positive cells following incubation of GFP-LINGO2 transfectants with PBS **l** IL-2-647, or **m** TFF3-647. **n** Double positive HEK following incubation of TLR2-GFP transfectants with TFF3-647. **o** with SDF-1-647 or **p** CXCR4-GFP transfectants with SDF-1-647. Data represent three independent experiments with mean ± SE and results of paired two-tailed *t* test shown. **q** Representative immunofluorescence images of rectosigmoid biopsy tissue samples from normal human subjects following co-staining with IgG isotype mAb or two different human subjects with anti-TFF3 mAb and anti-LINGO2 mAb (20×) (**r**, **s**). Scale bar represents 40 μm

TFF3-647 did not bind to other LRR receptors such as TLR2 (Fig. 1n) and SDF-1-647 did not bind LINGO2 (Fig. 1o). TFF3 showed moderate binding to CXCR4 transfectants (Fig. 1p).

Next, to determine whether endogenous TFF3 production and LINGO2 expression occurred in a similar intestinal microenvirononment to allow biologically relevant interactions in situ, normal human rectal tissues were immunostained with TFF3 and LINGO2 Ab to assess their spatial patterns. Compared to control Ab (Fig. 1q), LINGO2 was expressed in both enterocytes and sparsely along crypt epithelia, whereas TFF3 stained goblet cells located within crypts (Fig. 1r, s). In some instances, patches of cell-free TFF3 localized to the apical surface of LINGO2 positive enterocytes (Fig. 1r). Furthermore, distance between TFF3 and LINGO2 was determined using proximity ligation assay (PLA), which tests whether two Ab-bound proteins are within the 40 nm required for rolling circle amplification to produce fluorescent punctae[41]. Compared to single stained anti-LINGO2 mAb (Supplementary Fig. 1D and F), combined staining with TFF3 and LINGO2 Ab revealed puncta within intestinal lamina propria and crypt epithelia (Supplementary Fig. 1E and G). Collectively, these data suggested that TFF3-LINGO2 interactions occurred within 40 nm on the cell membrane.

**TFF3 requires LINGO2 to activate EGFR and STAT3.** TFF3-mediated cytoprotective functions are associated with EGFR and ERK activation[13,42,43,15]. To determine if TFF3 impacted epithelial EGFR signaling through LINGO2, we used a MC38 cells (murine intestinal adenocarcinoma) that expresses all four LINGO family member genes (Fig. 2a). A *Lingo2*KO subclone of MC38 (Δ-LINGO2) was generated using CRISPR-mediated insertion of a GFP-H2-K$^k$ knock in construct followed by FACS sorting (Fig. 2b) showing a 20-fold reduction in LINGO2 mRNA expression (Fig. 2c). To directly test whether TFF3 protected against cytotoxicity in a LINGO2-dependent manner, we compared the parental MC38 vs. Δ-LINGO2 MC38 lines in cell-death assays using the apoptotic agent, staurosporine[44]. Whereas TFF3-Fc pre-treatment significantly reduced cytotoxicity of parental MC38, protection was abrogated in Δ-LINGO2 cells (Fig. 2d). Given the anti-apoptotic role for STAT3 and evidence that TFF3 regulates STAT3 at the promoter level, we asked whether TFF3-driven STAT3 activation required LINGO2. TFF3-Fc (1ug/ml) induced STAT3 phosphorylation (pStat3 705) within 5 min in parental MC38 that slowly dissipated over 30 min (Supplementary Fig. 2A, B). However, TFF3-Fc-treated Δ-LINGO2 cells lacked pSTAT3 to this extent (0.5 fold after 5 min treatment) and showed reduced total STAT3 levels compared to loading controls (Supplementary Fig. 2A, B). Moreover, TFF3-Fc-induced pEGFR within parental MC38, but only a marginal pEGFR increase in Δ-LINGO2 cells (Fig. 2e). Thus, LINGO2

facilitated TFF3 mediated cytoprotection, STAT3 induction, and EGFR activation.

**LINGO2 binds to EGFR and TFF3 reverses that interaction.** Evidence suggesting that LINGO1 (a LINGO2 paralog) binds to and blocks EGFR activation[45] prompted us to investigate whether LINGO2 directly interacted with EGFR. Reciprocal co-immunoprecipitation experiments demonstrated that Flag-LINGO2 and GFP-EGFR bound to each other (Fig. 2f) and confocal microscopy experiments demonstrated that EGFR-GFP and LINGO2-RFP co-localized on co-transfected cells (Supplementary Fig. 2C). Because TFF3 required LINGO2 to induce pEGFR (Fig. 2e), LINGO2 could pull-down and co-localize with EGFR, and LINGO1 inhibits EGFR, we surmised that a constitutively bound LINGO2-EGFR complex may be disrupted by TFF3 binding to LINGO2 thereby sequestering LINGO2 away from EGFR to facilitate pEGFR activation. Consistent with this hypothesis, Δ-LINGO2 cells expressed levels of the EGFR-responsive genes *Areg* and *Egf*, that were 4–5-fold increased compared to parental MC38 following exposure to EGF (Fig. 2g, h). Furthermore EGF-treatment induced stronger pEGFR accumulation in Δ-LINGO2 cells compared to parental MC38 (Fig. 2i).

To directly test whether TFF3 disrupted a constitutively bound LINGO2-EGFR complex, we devised a strategy wherein LINGO2-EGFR protein complexes were co-incubated with either media, an irrelevant cytokine (rIL-2) or rTFF3 in solution. Complexes were immunoprecipitated to detect Flag-LINGO2 and immunoblotted for GFP to detect EGFR. Data show complex formation remained intact following media only or rIL-2 treatment, but rTFF3 reduced the abundance of LINGO2-EGFR complexes (Fig. 2j). Lastly, we noticed the doubling rate of Δ-LINGO2 cells was consistently higher than the MC38 parental line which, if our model was correct, relied upon constitutive pEGFR. Indeed, treatment with the EGFR tyrosine kinase inhibitor Gefitinib significantly reduced the doubling rate of Δ-LINGO2 cells over a three-day period, but had minimal impact on the parental MC38 line (Fig. 2k). Altogether, this indicated a model wherein an otherwise constitutive LINGO2-EGFR repressive complex was disrupted by TFF3, allowing EGFR activation through ligand dependent or independent mechanisms, explaining how TFF3 could drive EGFR activation without being an EGFR ligand[46].

**LINGO2 deficient mice have excess intestinal EGFR activation.** LINGO2 knockout mice (*Lingo2*KO) were generated through CRISPR/CAS9 gene-editing using gRNAs for gene disruption through non-homologous end joining (NHEJ). Of the several *Lingo2*KO founder lines created, we selected one strain with a

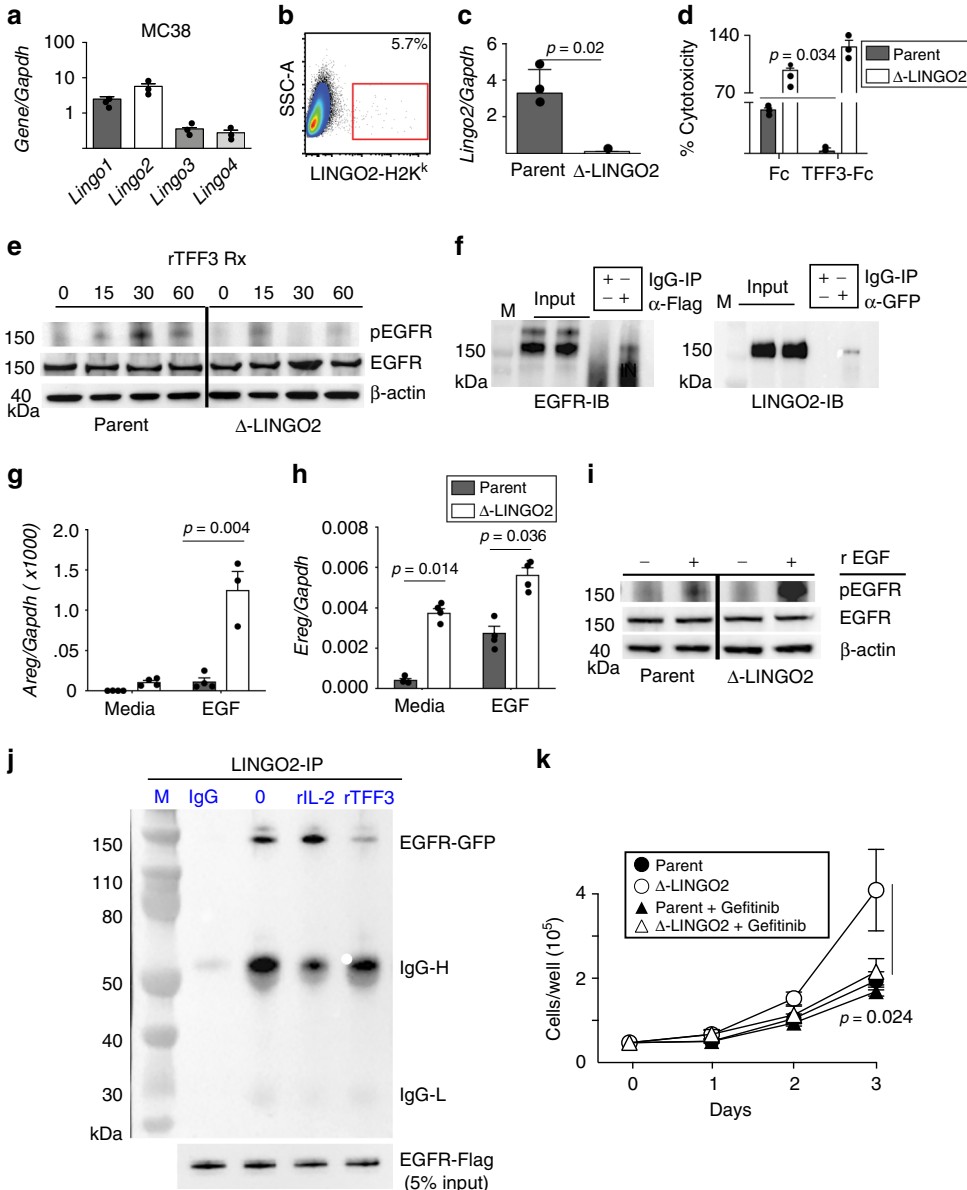

**Fig. 2** TFF3 requires LINGO2 to block cytotoxicity, regulate STAT3 and de-repress EGFR activation. **a** LINGO family mRNA transcript levels in the MC38 mouse intestinal cell line. Mean ± SE of 3–6 replicates are shown. **b** Gating strategy for FACS-sorting based isolation of CRISPR/CAS9 gene-edited MC38 cells via incorporation of the LINGO2-H2-K$^K$ knock-in construct designated as "Δ-LINGO2" **c** *Lingo2* mRNA transcript levels in Δ-LINGO2 MC38 cells as determined by QRT-PCR. Mean ± SE of triplicate wells. **d** Staurosporine-induced cytotoxicity at 6 h-post exposure of MC38 vs. Δ-LINGO2 cells following exposure to Fc (1 μg/mL) or TFF3-Fc (1 μg/mL or 23.81 nM) in DMEM 1% FBS. Mean ± SE of triplicate wells. **e** Time-course of phospo-EGFR (Y-1068) vs. total EGFR levels compared to beta-actin loading control in MC38 vs. Δ-LINGO2 cells exposed to Fc vs. TFF3-Fc (1 μg/mL). **f** Reciprocal co-immunoprecipitation of LINGO2-Flag and EGFR-GFP in co-transfected HEK293 cells. IP was performed with anti-Flag followed by IB with anti-GFP (left) or IP with anti-GFP followed by IB with anti-Flag (right). **g** mRNA expression levels of Amphiregulin (*Areg*) and **h** Epiregulin (*Ereg*) compared to *Gapdh* in parental MC38 vs Δ-LINGO2 cells following exposure to either media alone or rEGF (1 μg/mL) Mean ± SE of triplicate wells shown. **i** Phospho-EGFR (Y-1068) and total EGFR levels in parental MC38 vs Δ-LINGO2 at baseline or after stimulation with rEGF (1 μg/mL). **j** Co-IP of LINGO2-Flag and EGFR-GFP complexes in the presence of media only, soluble rIL-2, or rTFF3. **k** Kinetic analysis of cellular turnover for parental MC38 vs. Δ-LINGO2 cells in the presence or absence of the EGFR inhibitor Gefitinib (5 μM) treatment. Mean ± SE of six replicates are shown. Results of unpaired two-tailed *t* tests shown as *p* values in **c**, **d**, **g**, **h** and two-way ANOVa analyses in **k**

200 bp deletion immediately downstream of the transcriptional start site (Supplementary Fig. 3A). For initial characterization, intestinal tissues were subjected to LINGO2pAb immunostaining which revealed lack of LINGO2 on small intestinal villi in *Lingo2*KO mice compared to WT C57BL/6 controls and isotype matched Ab (Supplementary Fig. 3B–D).

To test our model that predicted LINGO2 constitutively suppressed EGFR, whole colonic tissues were isolated from WT

and *Lingo2*KO mice and several animals were compared for their total vs. activated pEGFR levels under naive, steady-state conditions. Colonic tissues from *Lingo2*KO mice had no difference in colon length vs WT (Supplementary Fig. 3E), but higher basal levels of pEGFR compared to WT tissue (Fig. 3a). Consistent with enhanced EGFR activity, despite macroscopically similar appearing colonic crypts and colon lengths (Fig. 3b), quantification of S-phase crypt epithelial cells

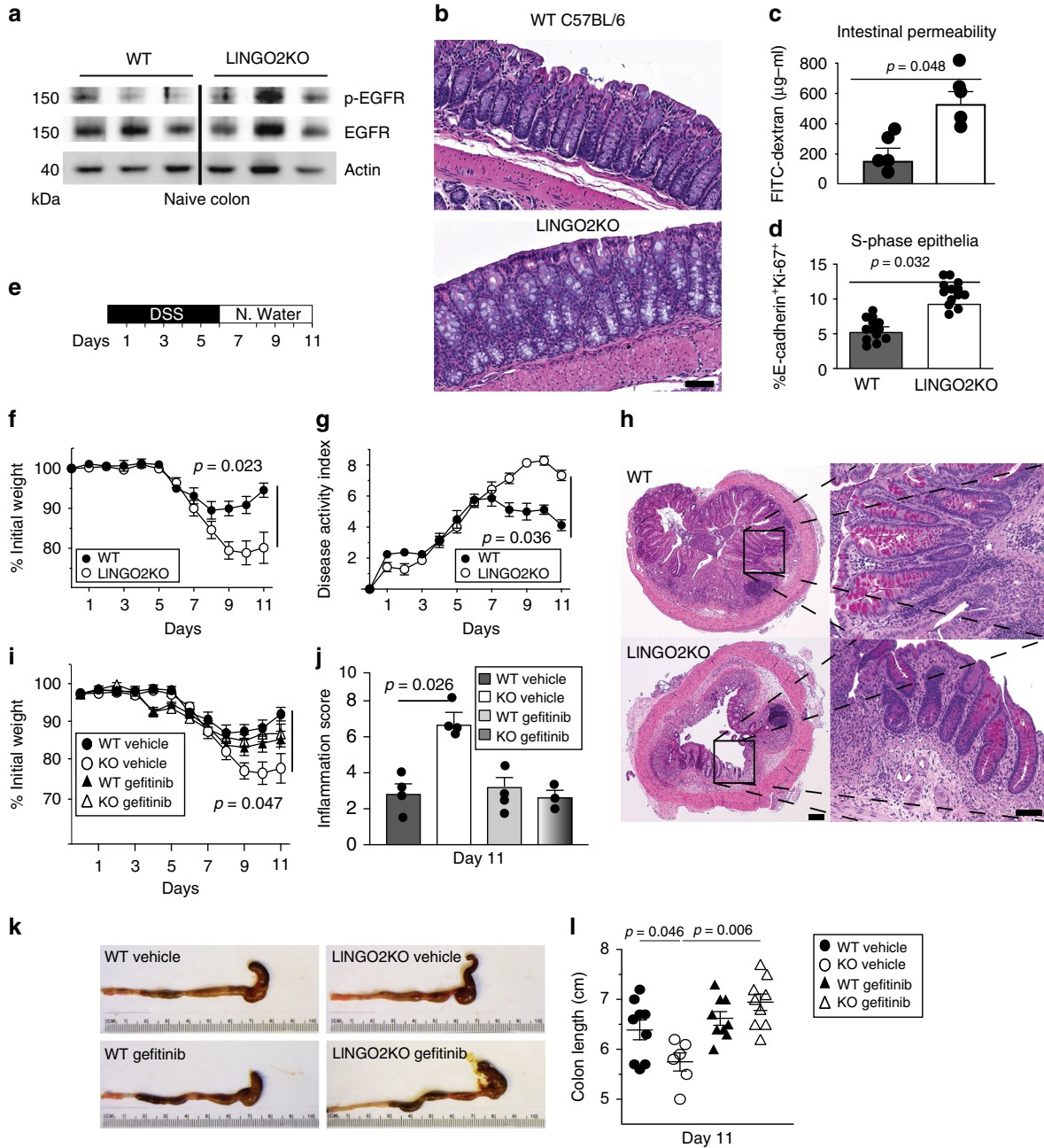

**Fig. 3** LINGO2/EGFR axis controls DSS colitis severity. **a** Western blot showing phospo-EGFR (Y-1068) vs. total EGFR levels compared to beta-actin loading control in small intestinal lysates from individual WT vs. *Lingo2*KO mice. Each lane is an individual mouse **b** H&E stained colon tissue from naive WT and *Lingo2*KO mice. Magnification ×40, scale bar 10 μm. **c** Serum levels of FITC-Dextran in WT vs *Lingo2*KO mice 4 h after oral gavage. **d** Percentage of small intestinal crypt cells per high power field that stained double positive for E-cadherin and Ki-67 in naive WT vs *Lingo2*KO mice. Mean ± SE of two independent experiments with the results of unpaired two tailed *t* tests shown. **e** Schematic of DSS treatment protocol. **f** Percent change from initial body weight caused by DSS administration at 2.5% w/v in cohorts of WT versus *Lingo2*KO mice. **g** Disease activity index (DAI) clinical scores in WT versus *Lingo2*KO mice from mice in **f**. Mean ± SE of 8–10 mice/group are shown. **h** H&E (left) and PAS (right) stain for WT and *Lingo2*KO rectum 11 days post 2.5% DSS treatment. Representative images are shown. Magnification ×4 (left) scale bar 100 μm and ×40 (right) scale bar 10 μm. **i** Change in initial body weight and **j** clinical inflammation score in WT versus *Lingo2*KO mice post 2.5% DSS with or without Gefitinib treatment. Means ± SE of 8, 9 or 10 mice/group are shown. **k, l** Colon length in WT versus *Lingo2*KO⁻ mice at day 11 post 2.5% DSS with or without Gefitinib. Means ± SE of 8, 9 or 10 mice/group and the results of two-way ANOVA analyses are shown

following anti-E-cadherin and anti-Ki-67 mAb co-immunostaining revealed a higher basal number of Ki-67 positive cells in the *Lingo2*KO compared to the WT (Fig. 3c). Also, *Lingo2*KO mice displayed impaired mucosal barrier function, demonstrated by 4-fold higher levels of FITC-dextran (4 kDa) in the peripheral blood at 4 h. post intragastric

administration as compared to similarly treated WT mice (Fig. 3d). However, 16srDNA sequencing to determine fecal microbiome composition did not reveal any substantial differences between strains (Supplementary Fig. 3F). Thus, in mice *Lingo2* deficiency resulted in heightened pEGFR activity and increased permeability of the GI tract at the steady-state.

**DSS-induced colitis in *Lingo2*KO mice is EGFR-dependent.** To determine the biological consequence of excess EGFR activation on intestinal injury and repair, *Lingo2*KO and co-housed WT littermates were orally administered dextran sodium sulfate (DSS) to induce transient colitic tissue injury (Fig. 3e). At 5 days following cessation of 2.5% DSS treatment, *Lingo2*KO mice had severe colitis marked by significantly greater weight loss and disease activity as compared to WT controls (Fig. 3f, g). Colon immunopathology was more severe in *Lingo2*KO mice, with greater degree of submucosal edema, ulceration, smooth muscle hyperproliferation and inflammatory infiltration as compared to WT controls (Fig. 3h). These experiments were then done with Gefitinib treatment to determine if inhibition of EGFR would reverse colitic disease in *Lingo2*KO mice. Indeed, weight loss, colon inflammation clinical scores, and colon shortening were all significantly reduced by Gefitinib treatment of *Lingo2*KO with negligible effect in the WT (Fig. 3i–l).

LINGO2 mRNA expression was detected in IEC, but also in myeloid cells, B cells, and CD4[+]T cells (Fig. 4a), therefore irradiation bone marrow chimeras were generated to test the contribution of hematopoietic vs. non hematopoietic cells (Supplementary Fig. 4A-D). Compared to WT-WT chimeras, both WT-KO and KO-WT groups showed worse colitis, as defined by weight loss, disease activity, and clinical pathology, yet neither was as severe as the KO-KO group (Fig. 4b–e). There were no differences in colon length (Supplementary Fig. 4E). Surprisingly, despite a report that *Tff3*KO mice are highly sensitive to (DSS)-induced colitis[47], we found that *Tff3*KO were not more susceptible to colitis than WT (Fig. 5a–c). This unexpected outcome was likely due to EGFR activity because Gefitinib treatment of *Tff3*KO mice, but not of WT mice, exacerbated colitic disease (Fig. 5d–g). Indeed, *Egf* mRNA transcripts were significantly higher in DSS-treated *Tff3*KO mice compared to WT or compared to either strain following Gefitinib treatment (Fig. 5h). Collectively, this suggested that EGFR activity, perhaps due to increased availability of EGFR ligand, protected *Tff3*KO mice from an otherwise increased susceptibility to DSS-induced colitis.

**LINGO2 deficiency protects against helminths via EGFR.** EGFR signaling drives host protective immune responses against different GI nematode species[48]. Goblet cells secrete substances toxic to GI helminths including Muc2, Muc5ac, and RELM-beta and also secrete immunoregulatory molecules such as TFF2[49–52]. Mice were inoculated with the infective larval stage (L3) of *Nippostrongylus brasiliensis* (*N.b.*). In this system, $L_3$ migrate from the skin to the lung, down the esophagus, and into the jejunum within 3d where they develop into fecund adults that are expelled between 9–12 days post-infection via suppression of Type 1 (IL-12,IFN-γ) and induction of Type 2 cytokine production (e.g., Areg, interleukins 4, 5, 9, 13, 25, and 33)[51,53] (Fig. 6a). *Tff3* mRNA jejunal transcripts peaked between 5–7d post infection (Fig. 6b). Importantly, *N.b.*-infected *Tff3*KO mice harbored significantly more adult worms than WT mice at 9d (Fig. 6c). Increased susceptibility in *Tff3*KO mice was associated with increased IFN-γ production from mesenteric lymph node cells, without differences in Type 2 cytokines (Fig. 6d). At baseline, prior to infection, there were no differences in CD3 stimulated Type 1 vs. Type 2 cytokines between naïve WT and *Lingo2*KO mice (Supplementary Fig. 4F). Strikingly, *N.b.*-infected *Lingo2*KO mice had early reduced worm numbers by d4 (Fig. 6e) and reduced fecal egg numbers compared to WT between d7–8 post-infection (Fig. 6f). This protective phenotype occurred regardless of co-housing or the sex of the animals. Curiously, Type 2 cytokines (IL-9 and IL-13) were significantly reduced in

*Lingo2*KO MLN cells compared to WT (Fig. 6g, h). Host resistance in *Lingo2*KO mice was primarily due to non-hematopoietic cells, as irradiation BM chimeras demonstrated that transfer of WT BM into KO mice resulted in significantly less fecal egg output as compared to the KO⁻WT chimeras (Fig. 6i). Consistent with the notion that enhanced EGFR activity occurred in the absence of LINGO2, we found higher amounts of pEGFR in intestinal homogenates of *Lingo2*KO compared to WT (Fig. 6j, k). As our model predicted, Gefitinib treatment during *N.b.* infection reversed resistance in *Lingo2*KO mice resulting in significantly higher intestinal worms compared to vehicle-treated controls (Fig. 6l). This indicated that *Lingo2* deficiency protected against helminth infection, at least in part, due to EGFR activity.

## Discussion

We report a pathway operating at the GI mucosal interface controlled by LINGO2, a leucine rich repeat and Ig domain containing receptor. Our data support a model wherein TFF3 produced by goblet cells can bind the LINGO2 extracellular domain (NH₂ terminal 350 AA) to de-repress inhibitory LINGO2-EGFR complexes, thereby allowing TFF3 to help maintain mucosal barrier integrity, suppress colitis, and promote immunity against GI helminths. TFF3-mediated disruption of LINGO2/EGFR complexes seems to operate similar to a rheostat tuning the extent of EGFR activity inasmuch as total LINGO2 deficiency was deleterious due to excessive basal EGFR activity, but loss of TFF3 was somewhat protective against colitis (Fig. 5d, e) and also impaired worm immunity. Inhibition of EGFR in *Lingo2*KO mice reversed the phenotype in models of colitis and helminth infection. It is likely this EGFR activity was due to activation by other EGF ligands (e.g., Areg, EGF) or TFF members (i.e., TFF2). While there are certainly other endogenous negative regulators of excessive other EGFR activity such as RALT, LRIG, SOCS4 and SOCS5[54–56], this is the first demonstration that: (1) TFF family members activate EGFR through de-repression, (2) LINGO2 regulates mucosal repair and immunity, and (3) TFF3 exerts its function(s), at least partially, through LINGO2.

LINGO2 is a 606aa protein with a large extracellular portion (ectodomain), a transmembrane domain, and a short cytoplasmic tail. LINGO2 belongs to the family of leucine-rich repeat and IgG-like domain proteins that includes LRIG1[57] that is known for ErbB receptor family regulation. LRIG1, a tumor suppressor and stem cell marker, induces proteasomal degradation of EGFR (and its other three ErbB family members) via polyubiquination[58]. LINGO1 is a component of a heterotrimeric signaling complex comprised of the Nogo-66 receptor and the p75 neurotrophin receptor that directly binds EGFR[45] and inhibits EGFR signaling and axon regeneration[59]. In oligodendrocytes, LINGO1 suppresses signaling of the ErbB2 receptor by preventing lipid raft translocation where it can be phosphorylated[60]. Similar to other leucine-rich repeat and Ig-like domain proteins, we observed an ErbB receptor family negative regulatory function for LINGO2. Our data suggest that LINGO2 binds with EGFR constitutively, suppressing its ability to undergo phosphorylation. Whether this occurs through catalytic inhibition or receptor downmodulation remains unclear. Whether LINGO2 operates in a signaling complex similar to LINGO1[59] is also unknown. Regardless, our data support a mechanism wherein upon injury, TFF3 gains access to LINGO2 in a manner requiring the NH₂ terminus, sequestering it away from EGFR, thereby raising the level of EGFR activation at times when the mucosa needs to repair tissue or expel pathogens.

Contrary to the initial report of DSS-treatment of the *Tff3*KO mouse strain[47] and in contrast to the prevailing notion that TFF3

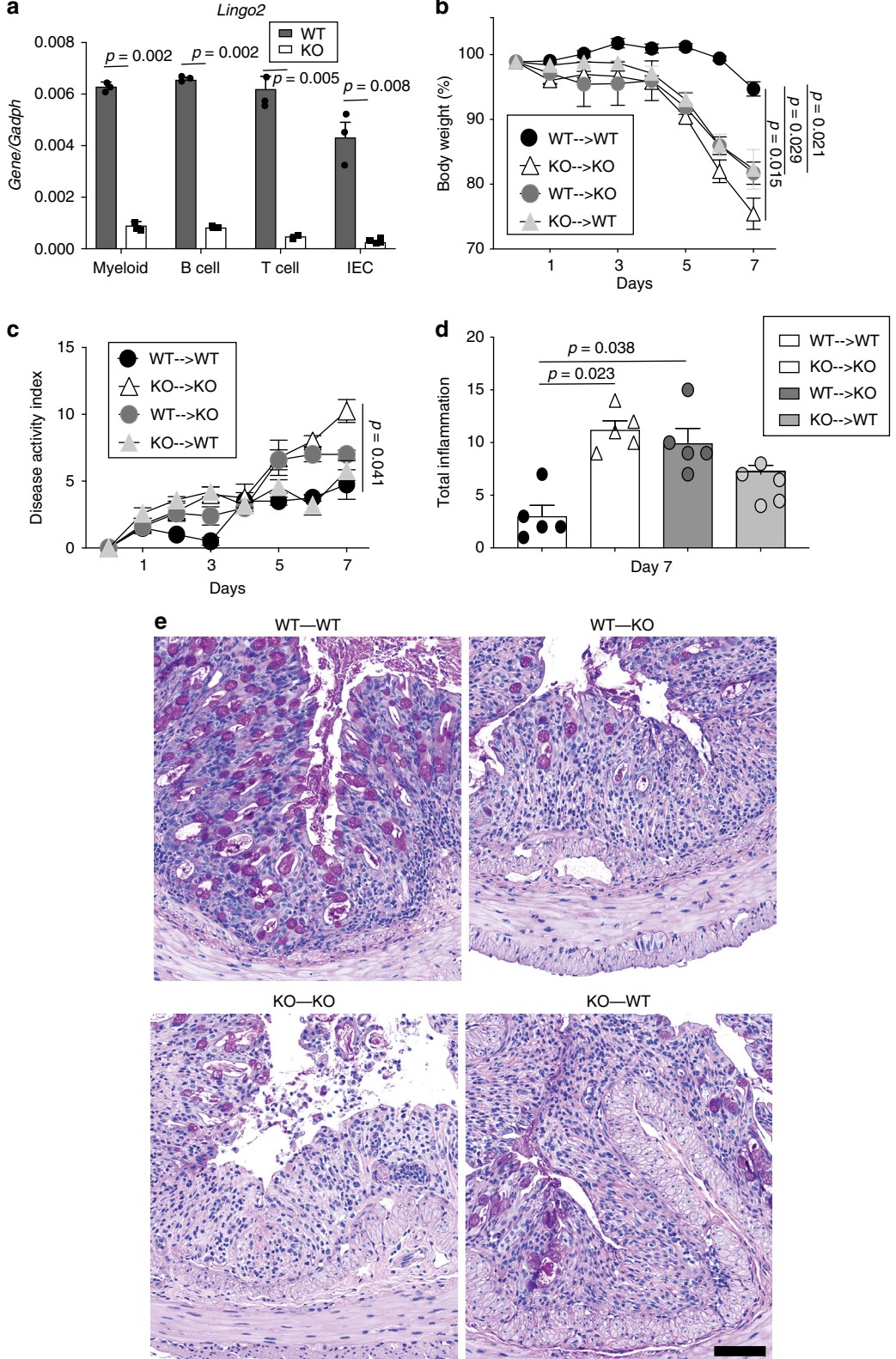

**Fig. 4** Non-hematopoietic and hematopoietic cells contribute to LINGO2-mediated protection from DSS colitis. **a** RT-PCR quantification of *Lingo2* mRNA transcript levels in myeloid, B cell, T cell and intestinal epithelial cells isolated using magnetic beads from WT and *Lingo2*KO mice $n = 3$/group. Representative of two experiments with results of unpaired two tail *t* tests shown. **b** Change in initial body weight and **c** disease activity index and **d** clinical pathology inflammation score in colon tissues from irradiation bone marrow chimeras at 4 days following cessation of 2.5% DSS administration for donor-recipient combinations WT → WT, *Lingo2*KO⁻ → *Lingo2*KO, WT → *Lingo2*KO, *Lingo2*KO⁻ → WT. Mean ± SE of 4 or 5 mice/group and results of two-way ANOVA analyses are shown. **e** Representative photomicrograph images at ×20 showing H&E stained colon tissues from irradiation bone marrow chimeras in (**b**–**d**). Scale 40 μm

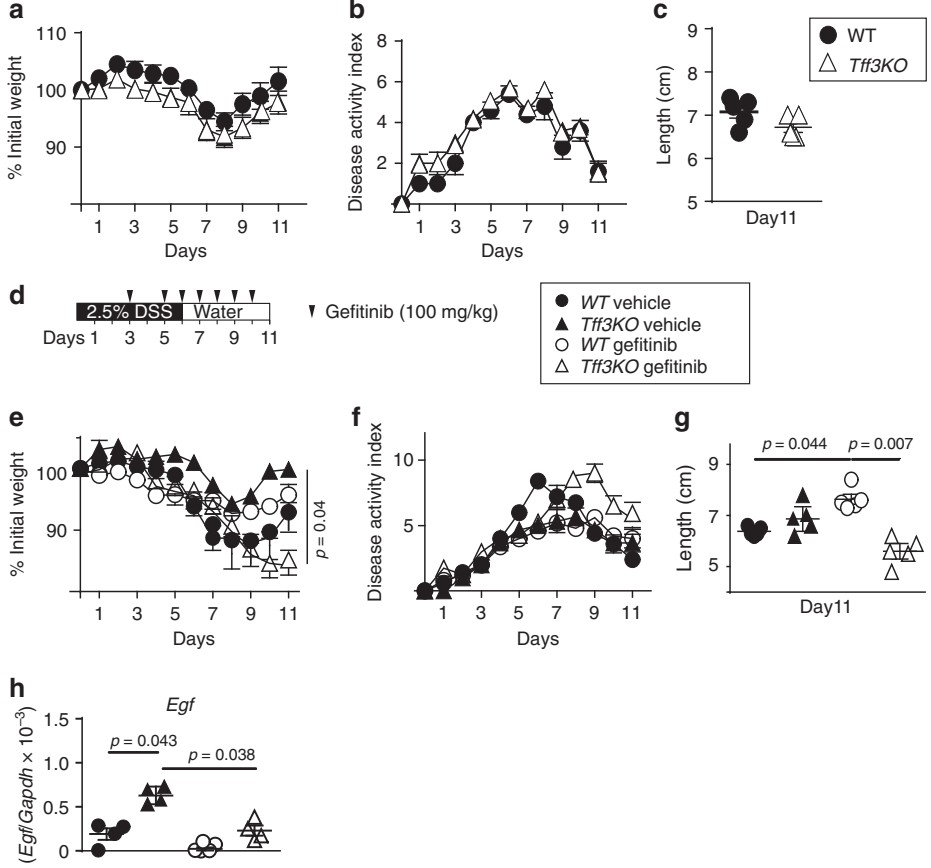

**Fig. 5** TFF3 deficiency in mice does not increase susceptibility to DSS colitis. **a** Change in initial body weight, **b** disease activity index scores and **c** colon lengths following DSS administration in cohorts of co-housed WT versus *Tff3*KO mice. **d** Gefitinib treatment protocol during DSS administration. **e** Change in initial body weight, **f** disease activity index, **g** colon lengths and **h** colonic *Egf* mRNA transcript levels at time of sacrifice in WT vs. *Tff3*KO cohoused mice following 2.5% DSS treatment with or without Gefitinib. Means ± SE of 8, 9 or 10 mice/group and results of two-way ANOVA analyses are shown

is essential for protection of the GI mucosa[61], we did not find that *Tff3*KO mice were more susceptible to DSS colitis. There may be several explanations for this discrepancy. The original study used mice on a mixed background (C57BL/6 × 129/SVJ), whereas our strain was backcrossed to C57BL/6 for >8 generations. Also, that report was published before recognition that microbial flora is a major determinant in GI inflammatory disease and did not take into consideration the importance of cohousing, whereas our studies were all completed using cohoused mice. Lastly, our protocol was 6 days on DSS followed by a return to normal drinking water in order to evaluate the wound healing response, whereas that study continuously treated mice with DSS for 9 days[47]. Notably, IEC-specific EGFR deletion in mice did not lead to any change in colitic disease severity indicating that normally the loss of EGFR on intestinal epithelia is not deleterious[62].

EGFR expression on Th2 cells is critical to their production of Type 2 cytokines that drive worm expulsion[31]. IEC also serve a host protective role against helminth infection[49,51,52]. In addition to secretion of toxic factors by IEC, an increased epithelial cell turnover provides host protection against helminths via an "elevator mechanism" that dislodges parasites from their niche through sloughing into the lumen[63]. *Lingo2*KO mice had higher basal numbers of proliferating epithelial cells in the crypt compared to WT, which when combined with the enhanced basal pEGFR activity, provides a plausible explanation for host resistance in *Lingo2*KO mice despite a reduced Type 2 cytokine response. Although there was no impact in the context of DSS colitis, *N.b.*-infected *Tff3*KO mice were more susceptible, which

was perhaps due to the increased IFN-γ production, known to block immunity against *N.b.* infection[64]. TFF3 may serve an additional role as an immunomodulatory cytokine similar to TFF2, which also suppresses Type 1 responses[65].

In conclusion, our results provide supportive evidence for a model that predicts LINGO2 binds to and inhibits EGFR activity and that TFF3 interactions with LINGO2 remove this negative regulation, allowing for enhanced EGFR activity. LINGO2 deficiency basally elevates EGFR activity, susceptibility to colitis and resistance to helminth infection. Genome wide association study (GWAS) data show LINGO2 genetic polymorphisms in human patients with cancer[66], asthma[67], chronic obstructive pulmonary disease[68], Parkinson's, and Inflammatory bowel disease[69]. In sum, this complex interplay among TFF3, LINGO2, and EGFR deserves greater attention to decipher how this triad controls mucosal repair and immunity.

## Methods

**Mice.** All animal procedures were approved by the Institutional Animal Care and Use Committee at University of California, San Francisco and at University of Pennsylvania. Super-ovulated female C57BL/6 mice (4 weeks old) were mated to C57BL/6 stud males. Fertilized zygotes were collected from oviducts and injected with Cas9 protein (30 ng/μl), sgRNA (15 ng/μl) into pronucleus of fertilized zygotes. Injected zygotes were implanted into oviducts of pseudo pregnant CD1 female mice. Animals were housed under specific-pathogen free barriers in vivarium at San Francisco General Hospital or University of Pennsylvania. Irradiation bone marrow chimeras were generated as previously described[70]. The recipients were irradiated with 10 Gy administered as a split-dose and were retro-orbitally inoculated with 2–5 × 10^6 total bone marrow cells isolated from the femurs of the donor animals within one hour following the second dose of irradiation. All irradiated recipients were administered 1% enrofloxacin for 2 weeks

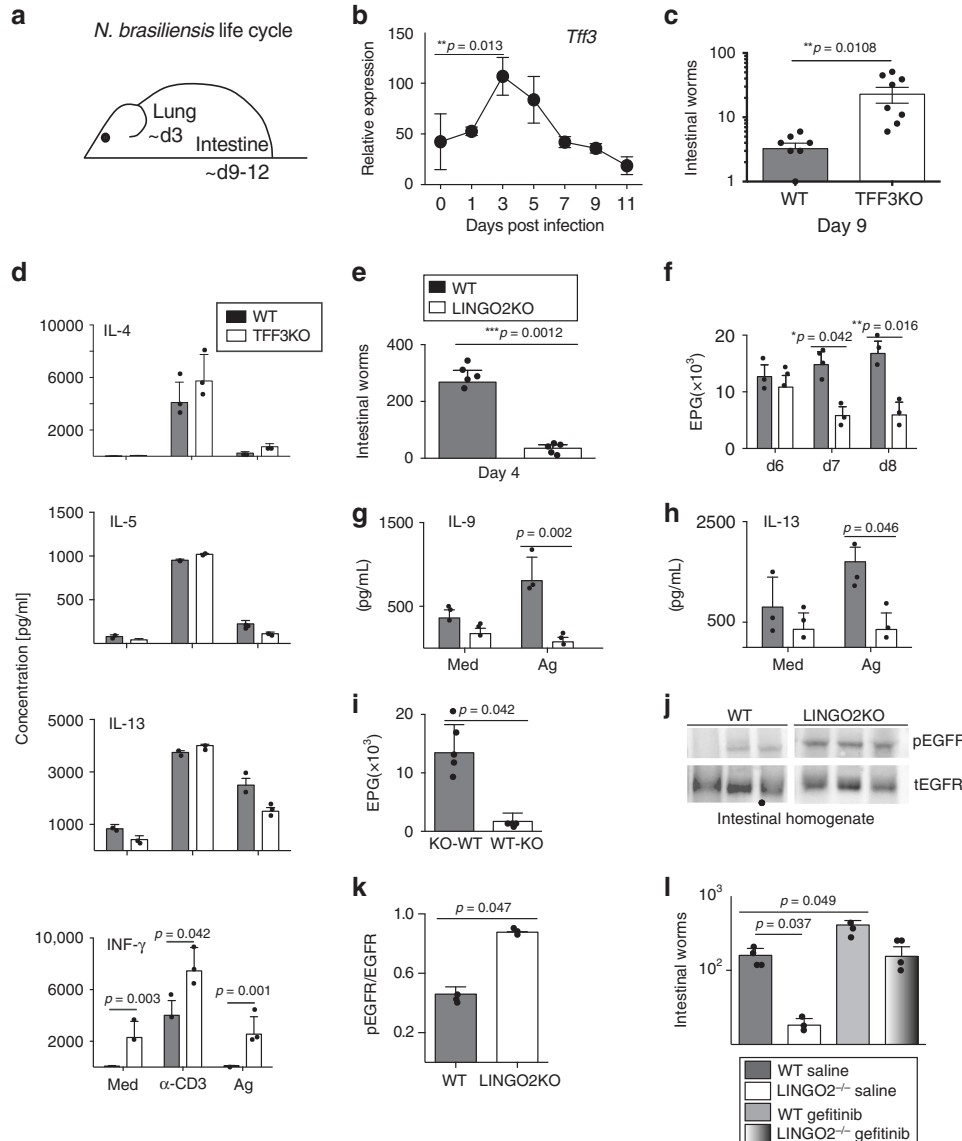

**Fig. 6** A TFF3-LINGO2-EGFR axis controls host-immunity against hookworm infection. **a** Diagram of *N. brasiliensis* infection life cycle in mice. **b** *Tff3* mRNA transcript levels in jejunum of WT C57BL/6 mice infected with 750 *N. b.* L$_3$ Data show mean ± SE of 3 or 4mice/group. **c** Intestinal adult worm numbers in WT vs. *Tff3*KO mice at day 9 post-infection each symbol represents an individual mouse. **d** Secretion of IL-4, IL-5, IL-13 and IFN-γ levels from WT vs. *Tff3*KO mesenteric lymph node cells isolated at day 9 post-infection. Cytokine levels in response to media alone, anti-CD3 (1 μg/ml or *N. b.* crude adult extract (10 μg/ml) at 48 h. Experiment performed twice $n = 4$–5 mice/group. **e** Intestinal worm numbers at day 4 post-infection 9 days post infection with *N. brasiliensis* in WT versus *Lingo2*KO mice. Mean ± SE of 5 mice/group are shown Representative of four independent experiments. **f** Fecal egg numbers, shown as eggs per gram feces (EPG) in WT vs. *Lingo2*KO mice at days indicated. **g** IL-9 and **h** IL-13 production from isolated MLN on day 9 exposed to media or *N. brasiliensis* antigen extract. **i** Egg load in feces on day 7 from WT → *Lingo2*KO versus *Lingo2*KO → WT BM chimera mice at day 4 post infection. Means ± SE of 3 or 4 mice/group are shown. **j** Western blot showing intestinal levels of phospho-EGFR and total EGFR in mice at day 9 post-infection. Each represents an individual mouse. **k** Densitometry for data shown in **j**. **l** Adult worm burdens recovered from intestinal lumen at day 9 from WT and *Lingo2*KO mice treated with Gefitinib or vehicle during infection with *N. b.* infection. Means ± SE of 9 or 10 mice/group are shown. *P* values shown represent the results of unpaired two tail *t* tests except for **l** where two-way ANOVA analyses are shown

post-irradiation and were evaluated for chimerism efficiency at 6–8 weeks post transfer. Only animals showing >90% donor chimerism were used for subsequent data analysis as determined by anti-CD45.1 and anti CD45.2 mAb combined with lineage specific Ab for CD4, B220 and CD11b and CD11c[70]. All the procedures complied with all relevant ethical regulations for animal testing and research and were reviewed and approved by IACUC at University of California at San Francisco (protocol #AN109782-01) and University of Pennsylvania (protocol #805911).

**DSS colitis and *Nippostrongylus brasiliensis* mouse models**. Colitis was induced in mice using 2.5% DSS (w/v, 0216011080, MP-Biomedicals LLC Solon, OH) in autoclaved drinking water for 6 days. Normal water was supplied on Day 6. Mice were assessed for body weight, appearance, fecal occult blood (Fisherbrand™ Sure-

Vue™ Fecal Occult Blood Slide Test System, Thermo Fisher Scientific, Inc., Waltham, MA, USA), stool consistency, and diarrhea during the entire course of colitis. Colon length was determined using a ruler at necropsy. Disease activity index (DAI) a cumulative score with a maximum of 12, was derived as follows: every mouse was assigned a score. 1 was for each display of lack of grooming, piloerection, awkward gait, hunched posture, and lack of mobility. Diarrhea score (0, Normal stool, 1 soft stool, 2 loose formed stools and 3, watery fecal matter), Rectal Bleeding was scored as: 0 no blood, 1 minor bleeding and 2 gross rectal bleeding). Fecal occult blood test was scored as: 0, Negative, 1 Mild amount of blood and 2 excessive presence of blood. Pharmacological inhibition of EGFR was achieved by administering Gefitinib (10 mg/Kg body weight i.p., G-4408 LC Laboratories, Woburn, MA) in DMSO. Gefitinib is an inhibitor of epidermal growth factor receptor (EGFR) tyrosine kinase domain that acts through binding to the

adenosine triphosphate (ATP)-binding site of the enzyme. Parasites were maintained on copro-cultures incubated at 25$^C$. Larvae from 7 to 14-day old cultures were collected and washed 3× in PBS with 1% Penicillin/Streptomycin. To infect with *Nippostrongylus brasiliensis*, 650–750 L$_3$ per mouse were injected subcutaneously using procedures previously established[51]. Mice were evaluated between d4–d9 for intestinal worm burden using the Baermann technique. On d6–d8 fecal pellets were collected from each mouse to assess egg counts. On designated days, mice were euthanized and the proximal half of small intestines were removed, opened longitudinally, and incubated in PBS for 2 h before collecting and counting adult worms.

**Microbial sequencing.** Genomic DNA was extracted from fecal pellets of mice using Qiagen DNeasy Power Soil Kit. The V4 region of the 16S rRNA gene was amplified using barcoded primers for use on the Illumina platform[32,33]. Sequencing was performed using 250-base paired-end chemistry on an Illumina MiSeq instrument. QIIME v. 1.8[34] was used to process paired-end reads and to perform downstream analyses including taxonomy assignment and heat map generation.

**Cloning/plasmid construction.** The full-length mLingo2 coding region was obtained with KpnI and XhoI double digestion of Lingo2 cDNA plasmid (MR215769, Origene) and subcloned into pcDNA3:GFP vector (Addgene) or pcDNA3:mRFP vector (Addgene) to generate pcDNA3:Lingo2:GFP and pcDNA3:Lingo2:RFP constructs. A serial of deletion of Lingo2 functional domains (Ecto and cytoplasm tails) were generated by PCR amplification with the specific primers bearing KpnI and XhoI overhangs, and cloned into pcDNA3:GFP vector (Adgene) to get Lingo2 truncated constructs fused with GFP. The full-length mTFF3 coding region was amplified using mTFF3 forward primer with HindIII overhang and mTFF3 reverse primer with XhoI overhang using mTFF3 cDNA plasmid (MR200151, Origene) as the template. The digested TFF3 PCR product with HindIII and Xho enzymes was cloned into pcDNA3:mRFP vector (Addgene) to generate pcDNA3:mTFF3:RFP construct. Using the similar method, the full-length mTFF3 coding region was also cloned into pINFUSE-mIgG2b-Fc2 vector (InvivoGen) to generate pINFUSE-TFF3:mIgG2b-Fc2 construct. The full-length mCXCR4 coding region was amplified with mCXCR4 forward primer with HindIII overhang and mCXCR4 reverse primer with XhoI overhang using mCXCR4 cDNA plasmid (OMu23034D, Origene) as the template. The digested CXCR4 PCR product with HindIII and Xho enzymes was cloned into pcDNA3:GFP vector (Addgene) to generate pcDNA3:mCXCR4:GFP construct. The full-length SDF1/CXCR12 coding region was amplified with mSDF1 forward primer with EcoRI overhang and mSDF-1 reverse primer with XhoI overhang using SDF1 cDNA plasmid (MR227229, Origene) as the template. The digested SDF1 PCR product with EcoRI and Xho enzymes was cloned into the pcDNA3:mRFP vector (Addgene) to generate pcDNA3:SDF1:RFP construct. The sequences of constructs were confirmed by DNA sequencing in UPenn Sequencing facility. HEK293 cells were maintained in DMEM medium (Invitrogen) supplemented with 10% fetal bovine serum (Invitrogen) and 1% penicillin-streptomycin (Invitrogen) at 25 °C with 5% CO2. Before transfection, 2 ml of cells (0.2 × 106/ml) per well was seeded on in a well of Lab-Tek Chamber slide (sigma) for overnight culture. 1ug plasmids Lingo2:GFP and TFF3:RFP were co-transfected to the cells using X-tremeGENE HP DNA transfection reagent (Roche) based on the manufacturer's manual. For co-localization of EGFR:GFP and Lingo2:RFP, pCMV3-mEGFR-GFPSpark (MG51091-ACG, Sino Biological Inc.) and pDNA3-Lingo2:RFP was transfected in the same method. After three-days culture, the cells were fixed with 4% paraformaldehyde in PBS for 10 min and stained with DAPI (0.5ug/ml) for 5 min. After mounting the cells with the coverslip, Lingo2:GFP and TFF3:RFP were visualized using the Leica TCS-NT confocal microscope.

**Human tissue and immunostaining.** Archived rectosigmoid biopsy samples were from the SCOPE cohort at the University of California, San Francisco (UCSF). The SCOPE cohort is an ongoing longitudinal study of over 1,500 HIV-infected and uninfected adults followed for research purposes. The UCSF Committee on Human Research reviewed and approved the SCOPE study (IRB# 10–01218), and all participants provided written informed consent. PLA starter Duolink® In Situ Orange Starter Kit Mouse/Goat was obtained from Sigma Aldrich containing the PLA probes anti-goat MINUS and anti-mouse PLUS with the detection reagent DuoLink® Orange. Fluorescence for proximity ligation was observed under a Cy3 filter on a Leica Inverted Microscope DMi8 S Platform Formalin fixed paraffin embedded (FFPE) tissues were stained with primary antibodies for anti-LINGO2 Ab was raised against Goat (R&D Systems; AF3679) and anti-TFF3 raised against mouse (eBioscience; 5uclnt3) were used to detect the LINGO2 and TFF3 interactions in situ and for PLAs that followed the manufacturers instruction (Sigma). Technical controls omitting either one of the antibodies served as a negative control as no circular DNA oligonucleotides were able to be amplified.

**Soluble ligand binding assay.** Purified murine IgG (RayBiotech), murine IL-2, murine recombinant TFF3 (Peprotech) and SDF1a (R&D systems) were labeled with Alexa Fluor 647 dye using the Alexa Fluor 647 protein labeling kit (A30009, Molecular Probes) per manufacturer's directions. HEK293 cells were maintained in DMEM medium (Invitrogen) supplemented with 10% fetal bovine serum

(Invitrogen) and 1% penicillin-streptomycin (Invitrogen) at 37 °C with 5% CO$_2$. HEK293 cells were seeded at $1 \times 10^6$/well and incubated at 37 °C with 5% CO$_2$ overnight before transfection with 3ug of plasmids pCMV3-mTLR2-GFP, pCMV3-mLINGO2-GFP or 6ug of pCMV3-mCXCR4-GFP using FuGENE HD DNA transfection reagent (Promega) based on manufacturer's instructions. 24 h post transfection, cells were incubated with either PBS or 3 µg of Alexa Fluor 647 labeled soluble ligands for 16 h at 4 °C. Cells expressing GFP and/or Alexa Fluor 647 positive were identified by flow cytometry using a LSRII.

**Co-immunoprecipitation.** HEK293 or CHO cells were transfected with 2 µg pCMV6:Lingo2:Flag (MR215769, Origene) plasmid. After 72 h., the cells were washed and lysed with 0.5 ml Digitonin extraction buffer [1% digitonin; 150 mM NaCl; 1 mM MgCl$_2$;10 mM TrisCl, pH 7.4, 1× Proteinase Inhibitor (Sigma)]. The protein concentration was measured using BCA assay kit (Invitrogen). A minimum of 0.6 mg total protein was incubated with 10 µg TFF3-Fc (Ray Biotech) or Fc (Ray Biotech) overnight and precipitated with 30 µl protein G agarose (Invitrogen) for 2 h at 4 °C. The IP complex was washed and eluted with 2× SDS loading buffer at 95 °C. The IP complex along with 5% input were immunoblotted with mouse anti-Flag antibody (1:5000) (Sigma) to detect the interaction between Lingo2:Flag and TFF3-Fc. In some cases, rabbit anti-FLAG followed by incubation with goat anti-rabbit IgG Cy5 and goat anti-mouse IgG Cy3 antibodies was used. To detect EGFR and Lingo2 interactions, HEK293 were co-transfected with 2 µg pEGFR:Flag (MG51091-CF, Sino Biological Inc.) and 2 µg pDNA3:Lingo2:GFP plasmids. Around 0.5 mg total protein was precipitated using 2 µg normal rabbit IgG (the negative control), 2 µg anti-GFP rabbit polyclonal antibody (Torrey Pines Biolabs), 2 µg normal mouse IgG (the negative control), 2ug anti-Flag mouse antibody (Sigma) and incubated with 30 µl protein A agarose (Invitrogen) for 2 h at 4 °C. The IP complex precipitated with anti-GFP antibody along with 5% input was immunoblotted with mouse anti-Flag antibody (1:5000) (Sigma) to detect the interaction between Lingo2:GFP and EGFR-Flag. The IP complex precipitated with anti-flag antibody along with 5% input was immunoblotted with rabbit anti-GFP antibody (1:5000) for the detection of reciprocal interaction. To determine if TFF3 disrupts the interaction between Lingo2 and EGFR, the mixture of 100ug Lingo2:GFP lysate and 50ug EGFR-flag lysate was incubated with either 2 µg TFF3 or 2 µg mIL2 (R& D System) proteins for two hours at 4 °C. Then, Co-IP was performed with either rabbit anti-GFP antibody or normal rabbit IgG as above. The IP complex was immunoblotted with mouse anti-Flag antibody (1:5000) (Sigma) to detect the interaction between Lingo2:GFP and EGFR-flag. IL-10 levels determined by ELISA according to the manufacturer's protocol (EBioscience).

**Cell lines.** The U-937 histocytic cell line was obtained from ATCC (ATCC® CRL-1593.2) and grown according to the suggested culture conditions in RPMI-10. Prior to stimulation with 100 ng/ml *E. coli* lipopolysaccharide, cells, were treated overnight with PMA to induce macrophage morphology. Supernatants were collected after 24 h in the presence or absence of rhTFF3 (Peprotech) for measurement of IL-10 or TNF using the manufacturer's suggestion. LINGO2 Knockout colon epithelial cell line (MC38) was generated based on HR (homologous recombination)-mediated CRISPR–Cas9 genome editing. Validated LINGO2 guide RNA obtained commercially (PNA Bio). Briefly, MC38 cells were maintained in DMEM medium (Invitrogen) supplemented with 10% fetal bovine serum (Invitrogen), 1XNEAA, 10 mM HEPES, 1 mM Sodium Pyruvate, 50ug/ml Gentamicin and 1% penicillin-streptomycin (Invitrogen) at 25 °C with 5% CO$_2$. Before transfection, cells ($1 \times 10^5$ per well) were seeded in six-well plates for overnight culture. 1ug Lingo2 sgRNA, 6 µg Lingo2 donor vector and 1ug Cas9 plasmid were co-transfected into the cells using X-tremeGENE HP DNA transfection reagent (Roche). After expanding the cells for two generations, H2-K$^K$ positive cells (Lingo2KO cells were identified by flow cytometry because Lingo2 donor vector contains a H2-K$^K$ gene cassette. Reduction of Lingo2 mRNA expression in H2-K$^K$ positive cells was confirmed by real-time QRT-PCR.

**Western blotting.** M38 parental or Lingo2 KO cell lines ($1 \times 10^5$ per well) were seeded in 24-well plates and cultured for 48 h. in DMEM-10. Cells were then washed and cultured in serum-free medium for 24 h. Cells were treated with TFF3-Fc (1ug/ml) for 0, 5 min, 15 min, 30 min, and 1 h and were lysed with buffer containing [1% Triton X-100, 150 mM NaCl,10 mM Tris (pH 7.4), 1 mM EDTA, protease and phosphatase inhibitor cocktail (Sigma)]. Mouse colon tissues from WT or Lingo2 KO mice were lysed with RIPA buffer containing protease and phosphatase inhibitor cocktail (Sigma). 15ug samples were loaded and separated in 4–12% PAGE gel (Invitrogen). The following primary Abs were used for immunoblotting assays: mouse anti-Stat3 (Cell signaling,1:5000), rabbit anti-pStat3 (Y705) (cell signaling, 1:100), rabbit anti-EGFR (Millerpore,1:1000), rabbit anti-pEGFR (Y1068) (Abcam,1:1000); mouse anti-actin (Santa Cruz, 1:2500). Uncropped blots are shown in the Source Data File.

**TRICEPS mass spectrometry.** The CaptiRec samples were analyzed on a Thermo LTQ Orbitrap XL spectrometer fitted with an electrospray ion source. The samples were measured in data dependent acquisition mode in a 40 min gradient using a 10 cm C18. The 12 individual samples in the CaptiRec dataset were analyzed with a statistical ANOVA model. This model assumes that the measurement error follows

Gaussian distribution and views individual features as replicates of a protein's abundance and explicitly accounts for this redundancy. It tests each protein for differential abundance in all pairwise comparisons of ligand and control samples and reports the *p*-values. Next, *p*-values are adjusted for multiple comparisons to control the experiment-wide false discovery rate (FDR). The adjusted p-value obtained for every protein is plotted against the magnitude of the fold enrichment between the two experimental conditions. The area in the volcano plot that is limited by an enrichment factor of twofold or greater and an FDR-adjusted p-value less than or equal to 0.01 is defined as the receptor candidate space. To allow for statistical analysis, the experiment was done in biochemical triplicates. The samples enriched in glycopeptides were analyzed on a Thermo LTQ Orbitrap XL spectrometer. Peptide identifications were filtered to a false-discovery rate of < =1% and quantified using an MS1-based label-free approach.

**Histology**. Mice were eviscerated, and colons extracted. A small biopsy of the intact distal colon was excised, fixed overnight at 4 °C in neutral 10% formaldehyde, further dehydrated and embedded in paraffin for histological analysis. Another small biopsy in the distal region was snap frozen in liquid $N_2$ prior to analysis. Remaining length of the colon was embedded as "swiss rolls" in OCT. Pathological assessment of colitis was performed blindly by certified pathologists as per established criteria for the Penn Vet Comparative pathology core.

Primers (Forward-Reverse)

mTTF3-5′ GCC CTC TGG CTA ATG CTG TT 3′ and 5′ TTG GGA TAC TGG AGT CAA AGC 3′

mLINGO1-5′ GAA CAA GAT CGT CAT CCT GC 3′and 5′ GAT GTT GAG ATG TCG TAG CC 3′

mLINGO2-5′ GCC CGT AAC CAA GGT GTA GAC-3′ and 5′ TTA AGA GTA ACA CCA CAG CCA GAC 3′

mLINGO3-5′ CTG CAC TTG CTG CTG CT 3′ and 5′ CTC CAG CAT GCG AGA TTC G 3′

mLINGO4-5′ AGA TGC AGC GCC CAA CTT GA 3′and 5′ CGG GAA TAG TGT CCA GTC G 3′

mGAPDH-5′ TGT GTC CGT CGT GGA TCT GA-3′and 5′ CCT GCT TCA CCA CCT TCT TGA 3′

Areg-5′ GCA GAT ACA TCG AGA ACC TGG 3′ and 5′ CTG CAA TCT TGG ATA GGT CCT 3′

Ereg-5′ CAC CGA GAA AGA AGG ATG GA 3′ and 5′ TCA CGG TTG TGC TGA TAA CTG 3′

mCXCR4-5′ CCA CCC AGG ACA GTG TGA CTC 3′ and 5′ GAT GGG ATT TCT GTA TGA GGA 3′

**RNA isolation and real-time qPCR**. Approximately 0.5 cm of duodenum was removed for RNA extraction using NucleoSpin RNA Plus kit (Macherey-Negel, Dueren, Germany). Five hundred nano-grams of total RNA were used to generate cDNA with Super Script II (Invitrogen). Quantitative real-time PCR was performed on the CFX96 platform (Bio-Rad, Hercules, CA). Gene expression levels were normalized to (*Gapdh*). Intestinal transcripts from colon biopsies were isolated and analyzed as previously described[51].

**Statistics**. Statistical analyses were performed using GraphPad Prism version 7.0 (GraphPad, La Jolla, CA). A two-tailed Student's *T* test or ANOVA was used where appropriate wth Kruskal Wallis used for non-parametric data (DAI scores).

**Reporting Summary**. Further information on research design is available in the Nature Research Reporting Summary linked to this article.

## Data availability
The authors declare that the data supporting the findings of this study are available from the authors upon request. The source data underlying Figs. 1a, c, d, 2a, c–k, 3a, c, d, f–g, i–j, l, 4a–d, 5a–l, 6b–l and Supplementary Fig. 2a–b is available as a Source Data file.

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

## Acknowledgements

We appreciate Anil Rustgi and Mike McCune for their critical evaluation of this manuscript.

## Author contributions

N.B., Y.J., Y.W., J.P., LY.H., K.Z., T.Y., K.H., T.O., C.P., S.S., and W.N. conducted experiments, M.S. contributed valuable reagents and samples, N.B., Y.J., Y.W., and J.P. helped to write the manuscript and D.R.H. conceived the study and wrote the manuscript.

## Competing interests

The authors declare no competing interests.
