## [Peer Review File · Nature Communications]

Reviewers' comments:

Reviewer #1 EGFR expert (Remarks to the Author):

The manuscript by Ji et al investigate the mechanisms by which the TFF3 cytokine drives reparative pathways of the gastrointestinal mucosa, in particular aiming at identifying a receptor for TFF3 and a mechanism of repair.

One of the main claims in this study are that LINGO2 functions as receptor for TFF3. The concern is that only indirect evidence of the involvement of LINGO2 as a TFF3 receptor is provided, including dose dependence reduction of endotoxin mediated TNF, the observation that glycosylation was critical for activity, the 8-fold enrichment of LINGO2 peptide using a TFF3 probe in Mass spec data, the co-precipitation of TFF3 and LINGO2 protein and TFF3-Fc and Flag-LINGO2, etc. These results consistently suggest interaction, but this does not need to be direct ligand-receptor interaction, but could rather be mediated by other proteins.

Also, the colocalisation of GFP-TFF3 and GFP-LINGO2 in transfected cells does not look convincing. Why were these cells fixed and permeabilised?

One would have liked to see direct experiments: For example, two-colour imaging of the receptor with a labelled mAb and labelled TFF3, showing not just colocalisation at low resolution, but for example FRET or super-resolution? Specific blocking of TFF3 binding by fluorescence mAbs?

A second claim is that LINGO2 is constitutively bound to EGFR in the steady state and that TFF3 binding to LINGO2 may sequester it from EGFR. This potentially can account for a repair mechanism.

Again only indirect proof is presented for this, like gene expression, and increased sensitivity in LINGO2 transfected cells to EGF-induced activation. Direct proof of this interaction is again missing. Fluorescence images in supplementary Fig. 2B show colocalisation at optical resolution. This however, does not prove a stoichiometric interaction.

I believe there are nowadays sufficient methodology in the literature to undertake a study that can prove direct interaction between molecules in cells. Co-immunoprecipitation and pull downs etc are not good enough for this. The images also have low resolution. The artefacts associated to colocalisation at low resolution are well understood (see for example Dunn et al, Am J Physiol Cell Physiol. 2011, and several others). Given that the conclusions rely on direct interactions between TFF3 and LINGO2 and LINGO2 and EGFR, direct proof should be presented for these.

Reviewer #2, expert in intestinal epithelium and response to helminths (Remarks to the Author):

This is a very nice piece of work from Ji et al, that presents data in favour of a novel signaling mechanism that can be relevant to mucosal immunity as determined by studies in DSS-induced colitis and *N. brasiliensis*-infected mice. This innovative work presents data in favour of TFF3 from goblet cells binding to LINGO2, that would constitutively be associated with the EGFR; the TFF3 causes dissociation of the LINGO2 thus de-repressing the EGFR and allowing signaling to proceed.

Using rodent and human tissues, transfected cell lines and murine models the authors present data that support the proposed signaling model. Collectively this is quite convincing. However, I have concerns with the robustness of the data, the completeness of the studies and numerous irksome errors in the paper detract from the overall message.

1. The robustness of many pieces of data is a concern. Why is there no indication of variability on

Figures 1A, 1C, 2G, S2A, S2B, S2C? There are no 'n' values in the figure legends in the supplementary material. There is no indication of how representative the images are of IHC or immunoblotting. The legend for figure 2 notes that n=3-6 replicates. If this is a single experiment, then have the data actually been reproduced? A number of statements are made on differences between groups but this is not always supported by statistical analysis (e.g. HB-EGF in S2C). Some of the data presented are non-parametric (i.e. disease activity index) and these cannot be analyzed by an ANOVA for parametric data. So while I like the novelty of the work, the question is are these data sufficiently solid to allow repetition and validate the conclusions stated.

2. The data with the DSS model are not clear. First, a small point, on figure 3 it would be useful to show the duration of the DSS treatment on panels A-C. What if any value do the 3% DSS data add to the paper? There are no absolute control data i.e. histology and colon length in a WT non-DSS treated mice or KO non-DSS treated mice. It is essential to have these data so that the reader can gauge the severity of the disease. Why is histopathology not scored, and statistics provided as for disease activity index (DAI)? In addition, it is disingenuous to omit the disease activity index and histopathology of the Gefitinib-treated mice. Mice can appear macroscopically well (based on DAI and colon length) and still have considerable histopathology. Since the goal in IBD is mucosal healing, it is pertinent for the reader to know if Gefitinib affects DSS-induced histopathology in either strain of mouse. Fundamentally, EGFR signaling would be expected to be good in DSS, and so the prediction with LINGO2 KO would be less, not more, DSS-induced disease. The authors present convincing data from the KO mouse but do not provide a conceptual frame to explain, or at least attempt to explain their findings. Finally, the microbiota are an important determinant of colitis in mice. To WT and KO mice have essentially the same colonic bacteria – if such data cannot be provided, the authors should at least acknowledge this.

3. The statements relating to some of the protein blots may not be defensible. In fig. 2E is the increased total STAT3 with the parent cells why there is more pSTAT3? The blot in fig.2F is not great quality and rather unconvincing (especially if it the best image the authors have). Does the increase total EGFR in Fig.2I LINGO2KO explain the increased pEGFR, i.e. is it simply relative expression related to total.

4. In figure 4, the data in panel D are all for infected WT or KO mice, what are the values in WT and KO non-infected mice? This is particularly important with the worm antigen, which, in theory, should be "0" in these mice. However, there can be mitogenic effects of worm preparations and it would be of value to know if the Ag effect is specific to infected animals. While not critical to this paper, data on worm infectivity at an earlier and a later time-point in the TFF3KO and LINGO2KO mice would be of value.

5. While the bulk of the manuscript seems tailored to convince the reader of the role of epithelial EGFR, the chimeric studies in figure 4 seem to point to a role for hematopoietic cells being important in the model. This is mentioned briefly and then skipped over. This seems to me like an important point worth exploring, at least with commentary in the text if not experimentation.

Minor

- Page 3: what is "gently oxidized" – not a scientific term.
- Figure S3B- the size of the scale bar on the panels is not provided.
- EPG, is presumably eggs per gram feces, but this is not defined in the figure legend or methods.
- Methods are missing, particularly relating to the *Nippostrongylus* study, eg. Cytokine measurements. There needs to a justification, or at least a reference, to support the kinetics of the model, Why this amount of DSS for 6 days, why this antibody treatment regime, why this does of Nb L3?
- N and P are missing in figure 1 legend
- Figure 2 legend needs to replace K and L with J and K

- Page 7 “Expressed higher levels of Areg and HB-Egf at day 9 compared to WT littermates”. Where are these data?

Reviewer #3, expert in intestinal epithelium and response to helminths (Remarks to the Author):

This study by Ji et al provides the first evidence for a specific receptor for the goblet cell-secreted protein TFF3. TFF3 is shown to interact with LINGO2 and indirectly promote EGFR signaling. LINGO2 knockout mice were generated and were shown to be highly susceptible to intestinal inflammation. This is a very interesting and important manuscript that provides new information.

1. It was unclear what was meant by the statement on p6, line 75. There are no data presented in the manuscript on TFF3 in DSS colitis. It would be very useful to include these data as 1) it would be good to know if the Mashimo et al. results are reproducible in your hands; and 2) it would be useful to compare severity of disease across multiple strains.
2. The DSS data in LINGO2 KO mice would be enhanced by using chimeric mice as in Fig 4I to show that the major effects of LINGO2 deletion are due to effects primarily on epithelial cells. It is hard to make any conclusions as the data presented in the manuscript show that many immune cells express LINGO2.
3. A description of Gefitinib for readers that may not be aware of how it functions would be helpful.

We appreciate the enthusiasm for our manuscript and now provide a point to point reply of all the revisions completed. The time for this revision was met with unforeseen difficulties as the previous first author became seriously ill and had to leave the laboratory unexpectedly. The revised manuscript has a new lead author, Nicole Maloney MD/PhD who was responsible for completing much of the requested experimentation. In this revision, we have included 2 entirely new figures in order to accommodate the data from the requested experiments. We have corrected the errors in the previous submission and have addressed each of the concerns raised in the initial review. We request the original reviewers of this manuscript.

Reviewers' comments:

Reviewer #1 EGFR expert (Remarks to the Author):

The manuscript by Ji et al investigate the mechanisms by which the TFF3 cytokine drives reparative pathways of the gastrointestinal mucosa, in particular aiming at identifying a receptor for TFF3 and a mechanism of repair.

One of the main claims in this study are that LINGO2 functions as receptor for TFF3. The concern is that only indirect evidence of the involvement of LINGO2 as a TFF3 receptor is provided, including dose dependence reduction of endotoxin mediated TNF, the observation that glycosylation was critical for activity, the 8-fold enrichment of LINGO2 peptide using a TFF3 probe in Mass spec data, the co-precipitation of TFF3 and LINGO2 protein and TFF3-Fc and Flag-LINGO2, etc. These results consistently suggest interaction, but this does not need to be direct ligand-receptor interaction, but could rather be mediated by other proteins.

Reply: We agree that while we have used multiple different methods to show that TFF3 and LINGO2 interact, these findings do not preclude the involvement of other transmembrane or intra-cellular proteins in a receptor complex. In fact, this is quite likely because LINGO1 is part of a tri-molecular complex that involves TROY (TNFRSF19) and RTN4R (Nogo-66/NgR). We respectfully request that the pursuit of LINGO2 co-receptors be deemed outside of scope for the present study. Therefore, our claims in the revised manuscript have been modified to indicate "Whether LINGO2 operates in a signaling complex similar to LINGO1 remains unknown"

Also, the co-localisation of GFP-TFF3 and GFP-LINGO2 in transfected cells does not look convincing. Why were these cells fixed and permeabilised? One would have liked to see direct experiments: For example, two-colour imaging of the receptor with a labelled mAb and labelled TFF3, showing not just colocalisation at low resolution, but for example FRET or super-resolution? Specific blocking of TFF3 binding by fluorescence mAbs?

Reply: The protocol that we had established required permeabilization and fixation. However, we agree that this approach may lead to intracellular interactions between LINGO2 and TFF3. Therefore, we have changed the approach in new experiments that did not require fixation or permeabilization that used intact

soluble Alexa Fluor 647 labeled TFF3 (TFF3-647) incubated with transfected HEK cells transfected with a vector encoding LINGO2-GFP followed by flow cytometry to identify doubly fluorescent cells. These new data (Figure 1K-P) show that in comparison to controls (PBS-LINGO2, IL-2-LINGO2, TFF3-TLR2 and SDF-1 LINGO2) there was at least a 4-fold increase in doubly fluorescent cells in the TFF3-LINGO2 group. This was a 12-fold increase as compared to interaction between IL-2-647 and LINGO2-GFP.

A second claim is that LINGO2 is constitutively bound to EGFR in the steady state and that TFF3 binding to LINGO2 may sequester it from EGFR. This potentially can account for a repair mechanism.

Again only indirect proof is presented for this, like gene expression, and increased sensitivity in LINGO2 transfected cells to EGF-induced activation. Direct proof of this interaction is again missing.

Reply: We respectfully disagree, as Figure 2J shows that incubation of Flag-LINGO2, GFP-EGFR protein complexes (migrating on the SDS page gel at 160 kDa) with rTFF3, but not rIL-2 results in a reduction in the abundance of this protein complex. We interpret this experiment as a specific disruption of LINGO2-EGFR complexes because the addition of rIL-2 which is of a similar molecular weight to rTFF3, does not lead to a disruption of this interaction implying that TFF3 interferes with an otherwise constitutive interaction between LINGO2 and EGFR.

Fluorescence images in supplementary Fig. 2B show co-localisation at optical resolution. This however, does not prove a stoichiometric interaction.

I believe there are nowadays sufficient methodology in the literature to undertake a study that can prove direct interaction between molecules in cells. Co-immunoprecipitation and pull downs etc are not good enough for this. The images also have low resolution. The artefacts associated to co-localisation at low resolution are well understood (see for example Dunn et al, Am J Physiol Cell Physiol. 2011, and several others). Given that the conclusions rely on direct interactions between TFF3 and LINGO2 and LINGO2 and EGFR, direct proof should be presented for these.

Reply: To directly address the reviewers concerns about direct interactions, we have included new experiments that describe proximity ligation assay (PLA) results. PLA is used to determine whether two proteins are within 40 nm on the cell membrane. We used antibodies specific for human LINGO2 and human TFF3 on human rectosigmoid tissue to induce a rolling circle amplification reaction that manifests as a fluorescent spot detected by fluorescence microscopy. The new data shows 100x magnification.

Reviewer #2, expert in intestinal epithelium and response to helminths (Remarks to the Author):

This is a very nice piece of work from Ji et al, that presents data in favour of a novel signaling mechanism that can be relevant to mucosal immunity as determined by studies in DSS-induced

colitis and *N. brasiliensis*-infected mice. This innovative work presents data in favour of TFF3 from goblet cells binding to LINGO2, that would constitutively be associated with the EGFR; the TFF3 causes dissociation of the LINGO2 thus de-repressing the EGFR and allowing signaling to proceed.

Using rodent and human tissues, transfected cell lines and murine models the authors present data that support the proposed signaling model. Collectively this is quite convincing. However, I have concerns with the robustness of the data, the completeness of the studies and numerous irksome errors in the paper detract from the overall message.

Reply: We appreciate the reviewer's comment that collectively our data constitute "a nice piece of work" in describing a novel interaction between LINGO2 and TFF3 and how this interaction regulates inflammation and immunity via EGFR signaling.

1. The robustness of many pieces of data is a concern. Why is there no indication of variability on Figures 1A, 1C, 2G, S2A, S2B, S2C? There are no 'n' values in the figure legends in the supplementary material. There is no indication of how representative the images are of IHC or immunoblotting. The legend for figure 2 notes that n=3-6 replicates. If this is a single experiment, then have the data actually been reproduced?

Reply: The data have been reproduced with two to three independent experiments. We initially included only a single experiment because cell lines were used and we did not pool the data, but now have done so in the revised version of the manuscript. We have also included n values in all our figure legends.

A number of statements are made on differences between groups but this is not always supported by statistical analysis (e.g. HB-EGF in S2C). Some of the data presented are non-parametric (i.e. disease activity index) and these cannot be analyzed by an ANOVA for parametric data. So while I like the novelty of the work, the question is are these data sufficiently solid to allow repetition and validate the conclusions stated.

Reply: We have changed the statistical analysis for the data presented and have chosen to use Kruskal Wallis for the DAI scores. Data in this resubmitted manuscript represents only data that have been analyzed for statistical significance and that have been re-produced at least twice.

2. The data with the DSS model are not clear. First, a small point, on figure 3 it would be useful to show the duration of the DSS treatment on panels A-C. What if any value do the 3% DSS data add to the paper?

Reply: We have now included a diagram showing the treatment protocol and have removed the 3% DSS treatment data in the revised manuscript. This approach of 6 days on DSS treated water followed by 5 days on normal water allows our experiments to focus on the repair phase (Figure 3E), which is where we find a clear importance for intact LINGO2 expression.

There are no absolute control data i.e. histology and colon length in a WT non-DSS treated mice or KO non-DSS treated mice. It is essential to have these data so that the reader can gauge the severity of the disease.

Reply: We apologize for this prior omission and now include data showing colon lengths for WT and LINGO2KO mice as well as clinical pathology scores for naïve irradiation chimeras under basal non injured conditions that were WT-WT, KO-KO, WT-KO, KO-WT donor – recipient combinations respectively. These data can be found in Supplemental Fig. 3E and 3F)

Why is histopathology not scored, and statistics provided as for disease activity index (DAI)? In addition, it is disingenuous to omit the disease activity index and histopathology of the Gefitinib-treated mice. Mice can appear macroscopically well (based on DAI and colon length) and still have considerable histopathology. Since the goal in IBD is mucosal healing, it is pertinent for the reader to know if Gefitinib affects DSS-induced histopathology in either strain of mouse. Fundamentally, EGFR signaling would be expected to be good in DSS, and so the prediction with LINGO2 KO would be less, not more, DSS-induced disease.

Reply:

Admittedly, the reviewer brings up an intuitive point that EGFR signaling should be expected to be good during DSS. Our data and also published data from a different group (Gastroenterology. 2017 Jul;153(1):178-190.e10.) indicates a counter-intuitive outcome, inasmuch as it is the level of EGFR signaling that matters most. In the case of LINGO2 deficiency the level of EGFR signaling is elevated such that the mucosal barrier integrity is compromised and DSS-induced colitic disease is exacerbated. We surmise that enhanced EGFR signaling particularly after DSS injury reduces barrier function of the newly produced IEC due to excessive proliferation resulting in IEC with a more immature phenotype. We show data supporting this contention for increased FITC dextran levels in the serum of LINGO2KO mice compared to WT following oral gavage (Fig. 3C) and also we found increased numbers of Ki-67+ crypt epithelia in LINGO2KO mice compared to WT (Fig 3D). Moreover, to address the reviewer's concerns, we now include histopathological clinical scores for the WT and LINGO2 chimeras in Figure 4D all of the groups included in Figure 3J and Figure 4C of the revised manuscript.

The authors present convincing data from the KO mouse but do not provide a conceptual frame to explain, or at least attempt to explain their findings. Finally, the microbiota are an important determinant of colitis in mice. Do WT and KO mice have essentially the same colonic bacteria – if such data cannot be provided, the authors should at least acknowledge this.

Reply: As mentioned above, we now provide a conceptual framework to explain the phenotype in the LINGO2KO and have included this explanation in the revised discussion section. In addition, we completed 16s rRNA microbial sequencing (paired end 250 on a Miseq) and analysis using QIIME1 to investigate

whether any possible commensal flora differences could explain the phenotype. New data included in the revised manuscript shows no consistent differences in microbial composition between the two strains.

3. The statements relating to some of the protein blots may not be defensible. In fig. 2E is the increased total STAT3 with the parent cells why there is more pSTAT3? The blot in fig.2F is not great quality and rather unconvincing (especially if it the best image the authors have).

Reply: We appreciate the reviewer's comment and concede that increased total STAT3 levels could explain this phenotype. Indeed, published work demonstrates that STAT3 is in a reciprocal inductive pathway with TFF3 production. We have revised our interpretation and included this highly relevant reference that describes the interdependence between STAT3 and TFF3 (Sci Rep. 2016 Jul 25;6:30421).

Does the increase total EGFR in Fig.2I LINGO2KO explain the increased pEGFR, i.e. is it simply relative expression related to total.

Reply: We appreciate the reviewers comment and concede that in some of our experiments it did appear as though total protein levels of EGFR were elevated in the absence of LINGO2, however this finding was not always consistent, therefore, we prefer the more conservative interpretation that pEGFR levels were primarily affected in the LINGO2 KO mouse.

4. In figure 4, the data in panel D are all for infected WT or KO mice, what are the values in WT and KO non-infected mice? This is particularly important with the worm antigen, which, in theory, should be "0" in these mice. However, there can be mitogenic effects of worm preparations and it would be of value to know if the Ag effect is specific to infected animals. While not critical to this paper, data on worm infectivity at an earlier and a later time-point in the TFF3KO and LINGO2KO mice would be of value.

Reply: We now include the baseline data as requested by the reviewer in Supplemental Fig 4 which shows no apparent differences between WT or LINGO2KO mice under basal naïve conditions. It does indeed look like the differences in Type 2 cytokines are evident only after infection.

5. While the bulk of the manuscript seems tailored to convince the reader of the role of epithelial EGFR, the chimeric studies in figure 4 seem to point to a role for hematopoietic cells being important in the model. This is mentioned briefly and then skipped over. This seems to me like an important point worth exploring, at least with commentary in the text if not experimentation.

Reply: We now include new commentary on the potential role for hematopoietic LINGO2 expression in the context of the irradiation chimera studies (Figure 4 B-E). This data demonstrates contributions from both haematopoietic and non-haematopoietic cells as both WT-KO and KO-WT groups had worse colitis than WT-WT but less than KO-KO cohort. We have now included discussion of how LINGO2 deficiency in the hematopoietic compartment likely has the opposite effect than deficiency in the non-hematopoietic

compartment, as it has been shown that EGFR signaling on CD4 T cells is necessary for host resistance to GI nematodes *Immunity*. 2017 Oct 17; 47(4): 710–722.e6.

Minor

- Page 3: what is “gently oxidized” – not a scientific term.

Reply: We agree and have replaced this phrase in the revised text with “PMA-treated U937 that had been treated with sodium periodate” in the revised manuscript.

- Figure S3B- the size of the scale bar on the panels is not provided.

Reply: We have now included a description of the scale bars on all images

- EPG, is presumably eggs per gram feces, but this is not defined in the figure legend or methods.

Reply: We agree and have now explicitly defined EPG as eggs per gram of feces in the legend of Figure 6.

- Methods are missing, particularly relating to the *Nippostrongylus* study, eg. Cytokine measurements. There needs to a justification, or at least a reference, to support the kinetics of the model,

Reply: We have extensive experience with the *Nippostrongylus* model and now include a revised methods section describing the infection and life cycle maintenance and evaluation of infected mice along with references of our previously published work.

Why this amount of DSS for 6 days, why this antibody treatment regime, why this dose of Nb L3?

Reply: We have included references of our previously published work using the DSS model. Because IBD is a relapsing remitting disease, we put mice on DSS for 6 days and return them to normal drinking water in order to study the healing (remitting phase). Concerning *N. b* infection, this is the maximal dose that elicits lung injury and a fulminant type 2 response without inducing mortality in the mice. This is also widely accepted in the field.

- N and P are missing in figure 1 legend

Reply: We have now revised all Figure legends throughout this manuscript and have corrected this error to include data description for all panels throughout.

- Figure 2 legend needs to replace K and L with J and K

Reply: We have now revised all Figure legends throughout this manuscript and have corrected this error to include data description for all panels throughout.

- Page 7 “Expressed higher levels of Areg and HB-Egf at day 9 compared to WT littermates”. Where are these data?

Reply: This was an error that has now been corrected. We now show elevated Areg and Ereg expression levels in Δ -LINGO2 cells as compared to the parental MC38 cell line in the revised manuscript.

Reviewer #3, expert in intestinal epithelium and response to helminths (Remarks to the Author):

This study by Ji et al provides the first evidence for a specific receptor for the goblet cell-secreted protein TFF3. TFF3 is shown to interact with LINGO2 and indirectly promote EGFR signaling. LINGO2 knockout mice were generated and were shown to be highly susceptible to intestinal inflammation. This is a very interesting and important manuscript that provides new information.

1. It was unclear what was meant by the statement on p6, line 75. There are no data presented in the manuscript on TFF3 in DSS colitis. It would be very useful to include these data as 1) it would be good to know if the Mashimo et al. results are reproducible in your hands; and 2) it would be useful to compare severity of disease across multiple strains.

Reply: Addressing this point has been a major part of our effort in the revision (see Figure 5). We have completed an extensive number of experiments designed to reproduce the Mashimo et al study. Unexpectedly, we found no support for an exacerbated phenotype in the DSS treated TFF3KO as compared to WT. However, upon EGFR blockade, we find that TFF3KO develop more severe colitic disease than similarly treated WT mice. We feel this is consistent with our overall model and actually would have been predicted by our model for the following reason. In the absence of TFF3, LINGO2 remains tightly bound to EGFR and restricts its over-activation, therefore TFF3KO mice are, if anything, more protected against DSS colitis than WT, and this effect is evident after crossing TFF3KO to RAG1KO mice Fig. 5 D-F. However, in the context of EGFR inhibition, LINGO2 serves no ability to protect. WT mice blocked of EGFR activity, show no enhancement of disease, which is consistent with data showing that loss of epithelial EGFR has no deleterious effect in the DSS colitis model. Gastroenterology. 2017 Jul;153(1):178-190.

2. The DSS data in LINGO2 KO mice would be enhanced by using chimeric mice as in Fig 4I to show that the major effects of LINGO2 deletion are due to effects primarily on epithelial cells. It is hard to make any conclusions as the data presented in the manuscript show that many immune cells express LINGO2.

Reply: We have completed these requested irradiation chimera experiments (Figure 4B-E). The data show that both hematopoietic and non-hematopoietic LINGO2 expression contributes to intestinal healing after DSS injury. There does seem to be a slight preference for the requirement of non-hematopoietic LINGO2, but both chimeric groups develop more severe colitis as compared to the WT-WT chimera. Also, see response to reviewer 2 above.

3. A description of Gefitinib for readers that may not be aware of how it functions would be helpful.

Reply: We have now included a description of the mode of action for Gefitinib in the revised manuscript on

in the methods section describing DSS treatment where it is stated, "Gefitinib is an inhibitor of epidermal growth factor receptor (EGFR) tyrosine kinase domain that acts through binding to the adenosine triphosphate (ATP)-binding site of the enzyme.

REVIEWERS' COMMENTS:

Reviewer #1 (Remarks to the Author):

I am happy with the response from the authors and have no further concern.

Reviewer #2 (Remarks to the Author):

No additional concerns- authors have responded satisfactorily to the issues that were raised.

Reviewer #3 (Remarks to the Author):

The authors addressed all my comments.

REVIEWERS' COMMENTS:

Reviewer #1 (Remarks to the Author):

I am happy with the response from the authors and have no further concern.

Reviewer #2 (Remarks to the Author):

No additional concerns- authors have responded satisfactorily to the issues that were raised.

Reviewer #3 (Remarks to the Author):

The authors addressed all my comments.

REPLY:
Great!